# Efficacy of Ultrasound for the Detection of Possible Fatty Liver Disease in Children

**DOI:** 10.3390/diagnostics14151652

**Published:** 2024-07-31

**Authors:** Sarah B. Lowry, Shelly Joseph, Kevin J. Psoter, Emily Dunn, Sana Mansoor, S. Kathryn Smith, Wikrom Karnsakul, Gihan Naguib, Kenneth Ng, Ann O. Scheimann

**Affiliations:** 1Johns Hopkins School of Medicine, Baltimore, MD 21205, USA; 2Langone Health Department of Pediatrics, New York University, New York, NY 10012, USA; 3Department of Pediatric Radiology, Johns Hopkins University, Baltimore, MD 21218, USA; 4Department of Pediatric Gastroenterology and Nutrition, The Herman and Walter Samuelson Children’s Hospital at Sinai, Baltimore, MD 21209, USA

**Keywords:** hepatomegaly, echogenicity, steatosis, NAFLD, MASLD, ultrasound, pediatric

## Abstract

Pediatric MASLD (previously referred to as NAFLD) incidence has continued to rise along with the obesity pandemic. Pediatric MASLD increases the risk of liver fibrosis and cirrhosis in adulthood. Early detection and intervention can prevent and reduce complications. Liver biopsy remains the gold standard for diagnosis, although imaging modalities are increasingly being used. We performed a retrospective study of 202 children seen in a pediatric gastroenterology clinic with a complaint of abdominal pain, elevated liver enzymes or MASLD, or a combination of the three to evaluate screening methods for MASLD. A total of 134 of the 202 patients included in the study underwent laboratory testing and abdominal ultrasound. Ultrasound images were reviewed with attention to liver size and echotexture by a fellowship-trained pediatric radiologist for liver size and echotexture. Overall, 76.2% of the initial radiology reports correctly identified hepatomegaly based on age and 75.4% of the initial radiology reports correctly described hepatic echogenicity that was consistent with increased hepatic fat deposition. Use of screening ultrasound in concert with other clinical evaluations can be helpful to identify children at risk of MASLD. Utilizing ranges for liver span according to age can help to diagnose hepatomegaly, and understanding how to identify hepatic echogenicity is important for identifying possible hepatic steatosis.

## 1. Introduction

Pediatric obesity rates have doubled over the past decade and quadrupled over the last four decades, and along with the increase in obesity, the rates of associated co-morbidities and diseases such as metabolic dysfunction-associated steatotic liver disease (MASLD), previously known as non-alcoholic fatty liver disease or NAFLD, diabetes mellitus, hyperlipidemia, and cardiovascular disease have also increased [1,2,3,4,5,6]. MASLD is chronic hepatic steatosis that is not associated with an underlying genetic or metabolic disease, with severity ranging from simple steatosis to steatohepatitis with fibrosis. MASLD is a leading cause of liver transplant in the adult population with the potential to progress to cirrhosis and hepatocellular carcinoma and requires early intervention to prevent progression of disease [2,3,4,7,8]. Disease progression can be reversible if detected early and managed with interventions, the most important being dietary changes and increased physical activity [2,3,8,9]. In the pediatric population, MASLD is the most common cause of chronic liver disease. Understanding how to best diagnose and monitor disease is important in order to intervene early in the course of the disease, to decrease risk of morbidity and mortality. The North American Society for Pediatric Gastroenterology, Hepatology and Nutrition (NASPGHAN) has put forth recommendations for screening for MASLD in pediatric patients with a focus on patients who are obese or have a risk factor associated with metabolic syndrome [3,4,10,11].

Imaging modalities are a useful tool to identify a normal anatomy, including liver and spleen size [12,13,14]. Non-invasive ultrasound (US) is the most utilized imaging modality for the detection of both hepatomegaly and fat infiltration in the liver [9,14]. The appearance of the liver is compared to the right kidney parenchyma to look for evidence of fatty infiltration, with sensitivity increasing as fat deposition percentages progress above 30% [3,7,14]. Hepatomegaly is defined as an increased liver span size based on normal ranges according to age [15]. Knowing these ranges is crucial for the appropriate detection of abnormal liver size in pediatric patients [7,13,15].

Previous studies have investigated ultrasound as a comparison to liver biopsy or magnetic resonance imaging (MRI) for fat detection through the evaluation of echogenicity [7,16,17]. There have been very few studies reporting on the prevalence of hepatomegaly and increased hepatic echogenicity detected by ultrasound in children [18]. Understanding this prevalence can lead to a timelier diagnosis of pediatric MASLD along with the opportunity to provide earlier intervention. A comparison of ultrasounds completed at ambulatory imaging centers to those performed at academic centers has also not been investigated previously. The primary aims of this study were to determine the prevalence of hepatomegaly and increased hepatic echogenicity diagnosed with ultrasound in pediatric patients and to compare the diagnostic accuracy of these findings when compared to an overread performed by a fellowship-trained pediatric radiologist.

This paper will review the methods with the study design, and the demographic details will be outlined in the results reported afterwards. Statistical analysis of the comparison of ultrasound reports as well as the determination of patients with a hepatomegaly diagnosis based on screening with presenting symptoms and complains will be reviewed. Finally, conclusions with impacts on patients and providers will be discussed in detail.

## 2. Materials and Method

### 2.1. Study Population

A retrospective study of children seen at Johns Hopkins Children’s Center, Mount Washington Pediatric Hospital (Weight Smart Clinic), and the Sinai Hospital of Baltimore between 2015 and 2018 was performed. Children were identified using ICD-9 and ICD-10 codes for diagnoses of abdominal pain, elevated liver enzymes, transaminitis, MASLD, fatty liver, and hepatomegaly. Children were included if they were between the ages of 2 and 18 years old, evaluated by a pediatric gastroenterologist, and received an ultrasound of the abdomen or right upper quadrant. For children who had more than one ultrasound, only the first clinical encounter in which an ultrasound was performed was included. Children who had an ultrasound that was unavailable for visual review were excluded. The institutional review board at the Johns Hopkins University School of Medicine approved the study.

### 2.2. Study Procedures

Demographics and clinical characteristics, laboratory results, and ultrasound reports were abstracted from the electronic medical record (EMR). Demographic and clinical data included presenting diagnosis, age at ultrasound, gender, race, height (cm), weight (kg), and body mass index (BMI). Laboratory assessments included a hepatic function panel, including electrolytes, transaminase levels, alkaline phosphatase total and direct bilirubin, albumin, and total protein. The classification of alanine aminotransferase (ALT) was based on the upper limit of normal (ULN) values, which is 22 U/L in females and 25 U/L in males [13]. ALT was considered elevated if it was greater than 2 times the ULN based on gender.

For ultrasound reports, reading and impression documentation recorded in the EMR were reviewed, as well as the training of the radiologist reading the ultrasound (adult or pediatric), which included liver span size and characteristics. All ultrasounds, available in Johns Hopkins University EMRs and on CRISP (Chesapeake Regional Information System for Our Patients), the shared medical record system amongst state hospitals in Maryland, were then reviewed by a fellowship-trained and board-certified pediatric radiologist who was masked to the initial ultrasound report. An overread was completed at a separate time and compared to the initial reading. These readings included liver size with suspicion of hepatomegaly, splenomegaly if present, and description of echogenicity. Liver size was interpreted using the standard ranges based on age in months, as part of the standard practice of care [15]. Liver texture and appearance were compared with other the right kidney on the ultrasound to determine the presence of heterogenicity or increased echogenicity, which is indicative of fat accumulation [5,7,14].

### 2.3. Outcomes

Hepatomegaly was defined as liver size greater than the ULN according to ranges based on age in months per standard references [15]. A description of the liver texture and/or echogenicity was provided, with heterogenous echogenicity or increased echogenicity used as a marker for fat deposition or accumulation suspicious for MASLD.

### 2.4. Statistical Analysis

Demographic and clinical characteristics of children were summarized and compared between children with and without hepatomegaly on ultrasound, and between children with and without increased echogenicity on ultrasound using Student *t* tests with unequal variances for continuous variables and chi-squared or Fisher exact tests for categorical variables. Prevalence of hepatomegaly and increased hepatic echogenicity on ultrasound were determined based on overreads performed by a fellowship-trained pediatric radiologist.

The overall agreement and diagnostic accuracy of the initial ultrasound reports were compared to overreads performed by the fellowship-trained pediatric radiologist who served as the gold standard. Summaries of diagnostic accuracy measures included sensitivity, specificity, positive predictive value (PPV), and negative predictive value (NPV). Agreement and diagnostic accuracy were determined for subgroups defined by patient gender, race, and ethnicity (White/Caucasian, Black/African American, Hispanic, Asian, Mixed Race, or Other), BMI z score categories (<−1.2, −1.2–1.5, >1.5), and radiologist training. A *p* value < 0.05 was considered statistically significant. All analyses were performed using STATA Version 16.1 (StataCorp, College Station, TX, USA).

## 3. Results

### 3.1. Study Population

A total of 202 children met the inclusion criteria based on demographics and laboratory findings, of which 134 (66%) children had ultrasounds available for review by the study’s fellowship-trained pediatric radiologist and comprised the study population. The most common diagnoses for indication for ultrasound were abdominal pain in 62 (46.3%) children and hepatomegaly on the physical exam in 50 children, with 32 of these children having increased liver size for age on ultrasound.

The characteristics of the study population are presented in Table 1. The mean age at time of ultrasound was 141.2 (SD = 53.45) months (or approximately 11 years and 9 months), and the majority of children was male (57.5%). White/Caucasian children comprised 34.3% of the study population, followed by Black/African American (32.1%) and Hispanic (20.9%) children. The mean BMI z-score was 1.27 (SD = 1.59) and the average ALT was 72.7 U/L (SD = 140).

### 3.2. Ultrasound Findings

Comparisons of characteristics of children with and without hepatomegaly are presented in Table A1. Elevated ALT was found in 75 (56.1%) of children, with a greater proportion of children with hepatomegaly on ultrasound having an elevated ALT compared to children without hepatomegaly on ultrasound (70.3% vs. 37.9%; *p* < 0.001). More than 50% of children who were White/Caucasian, Black/African American, and Hispanic had hepatomegaly on ultrasound, although the distribution of race did not differ significantly between children with and without hepatomegaly on ultrasound (Table 2).

The prevalence of increased echogenicity was 46.3% (95% CI: 37.6, 55.1%). Comparisons of characteristics of children with and without increased hepatic echogenicity on ultrasound are presented in Table A2. A greater proportion of children with increased hepatic echogenicity on ultrasound had an elevated ALT compared to children with normal hepatic echogenicity on ultrasound (77.1% vs. 38.0%; *p* value < 0.001). Increased hepatic echogenicity on ultrasound was seen in more than one third of all children for each race/ethnicity. Full details can be seen in Table 3.

### 3.3. Comparison of Initial Radiology Read to Read Performed by Fellowship-Trained Pediatric Radiologist for Hepatomegaly

The comparison of the initial radiology reading to that of the overread performed by the fellowship-trained pediatric radiologist regarding hepatomegaly is provided in Table 4. The overall agreement was 76.1% between the initial read and overread, with 19.4% of the original readings classified as false negatives, indicating the presence of hepatomegaly without a diagnosis in the original reading. Overall sensitivity was 64.9%, with a specificity of 90.0%, and a PPV and NPV of 88.9% and 67.5%, respectively.

Amongst subgroups of children based on demographic and clinical characteristics, the agreement ranged from a low of 69.3% in children with a high BMI z score to 100% in children with a low BMI z score. For males, the sensitivity was 75.5%, with a positive predictive value (PPV) of 88.1%. Amongst the race subgroups, an agreement between readings ranged from 88.2% in other racial groups to 69.6% in the White/Caucasian group.

Table 5 provides the details of the comparison of the initial radiologist’s read to the overreads performed by the fellowship-trained pediatric radiologist for the detection of increased hepatic echogenicity. Overall, the comparison between initial readings and overreads was 75.4% in agreement for the description of hepatic echogenicity. This suggests that the interpretation of echogenicity was based on experience. The false-negative rate of detection of increased echogenicity was 20.2%, meaning there were signs of possible fat deposition that was seen in the overread but not reported upon in the initial reading. Overall, the sensitivity was 56.5%, with a specificity of 91.7% a PPV of 85.4%, and NPV of 71%.

Within the subgroups of children, the agreement ranged from a low of 64.7% in Other race to 69.8% in the Black/African American group, to a high agreement of 84.8% in the White/Caucasian group. For females, the specificity was 94.4% with an NPV of 73.9%. Amongst the BMI z scores, there was an agreement in children with a BMI z score ranging from −1.2 to 1.5 of 71.7% to 100% compared to children with a BMI z score < −1.2.

## 4. Discussion

MASLD is defined as hepatic steatosis that is not related to a genetic or metabolic disorder and associated with hepatomegaly with fat deposition in the liver, and often has elevated liver enzymes. In this study, the prevalence of hepatomegaly and increased echogenicity detected using ultrasound in children were determined. The results demonstrate that ultrasounds in children provide a useful and accurate screening method for MASLD that is superior to a reliance on blood tests alone. The differences in laboratory values of ALT in children with and without hepatomegaly as well as in children with and without increased hepatic echogenicity were compared. Analysis of demographics also revealed a higher prevalence of increased hepatic echogenicity and hepatomegaly in Black/African American children than previously reported.

A physical examination is often the first tool used to assess for and identify hepatomegaly in children. Clinically, in patients who are obese with larger body habitus, it is more difficult to accurately identify hepatomegaly. Primary care physicians are often the first provider responsible for identifying children that require further evaluation and referral to pediatric gastroenterology. Especially in obese children, the presence of hepatomegaly is not easily identified in a physical exam, which often necessitates obtaining imaging, most often ultrasound, and labs, including a hepatic function panel [19].

The use of pediatric ultrasound for the recognition and monitoring of MASLD remains a common and frequently utilized tool due to its low cost, minimal risk, and wide availability [9]. About 75% of the time, the radiologist can determine the appropriate liver size based on age and changes in hepatic echotexture concerning fat deposition. For clinical practice, imaging, although frequently used, is still not a standard for diagnosis and work-up, which may lead to missed diagnoses. Early diagnosis remains crucial for changing the course of a disease [14].

ALT has been used as a screening test for MASLD and is the recommended first laboratory test for screening [20]. This is an easily obtained blood sample that is readily available worldwide. Currently, there is no universal standard for normal values and rates vary amongst different laboratories. Investigations into universal ranges based on gender have been performed [11]. However, ALT can have considerable ethnic variations as well, with Black/African American children having the lowest ALT values and Hispanic children have the highest ALT values [21]. As a result, Black/African American children are less likely to have elevated ALT values and have always been suspected to have a lower risk of development of MASLD [20]. Of greater concern, elevated ALT values do not always correlate with hepatic echotexture on imaging modalities since there may be steatosis without steatohepatitis present, and pediatric patients with normal ALT values can often still have increased hepatic echogenicity on ultrasound consistent with MASLD [21].

ALT values are an important screening tool for identifying patients at risk of MASLD, however these results show that, with the addition of ultrasound, the detection of at-risk children increases, which can impact clinical practice [20]. ALT can also be affected by several different disease processes, including viral illness, and may not always be the most accurate measurement of liver health. A consideration of appropriate ranges based on age remains crucial when screening using ALT, especially when considering the significant ethnic differences seen for normal values amongst Black/African American, Caucasian, and Hispanic pediatric patients [11,21].

Based on genetic factors, it was originally thought that there are ethnic differences when it comes to MASLD, with the disease being more prevalent in Hispanic and Caucasian individuals [4]. Male predominance has been shown as well in several epidemiologic studies and reviews [1,2,4]. Black/African American children have higher rates of insulin resistance and diabetes within the obesity epidemic but may not be monitored closely for MALSD due to the idea that they carry genes that protect against hepatic fat accumulation [4]. Interestingly, in this study, more than 50% of Black/African American patients presented with hepatomegaly and a higher percentage of Black/African American children had increased echogenicity, suggesting the possibility of undetected MASLD. It is important that this population of Black/African American children, with higher risk factors, is not overlooked based on previous identified genetic factors [4,6].

Apart from ultrasound, other imaging modalities are also helpful for the detection of hepatomegaly or the detection of changes in echogenicity. MRI and computerized tomography (CT) scans can both detect changes in the texture of the liver [7,10]. Despite these imaging modalities being more sensitive to identify changes early on in MASLD, they come with higher costs and increased risks. Hepatic proton-density fat fraction (PDFF) can be estimated by MRI as well and used as a marker for steatosis. Previous studies have showed mixed results regarding accuracy in different stages of fibrosis with ultrasound. PDFF can be more accurate than ultrasound alone at evaluating hepatic steatosis, but comes with the disadvantages of cost, accessibility, and need for sedation in younger children [22]. The gold standard for the diagnosis of MASLD remains to be liver biopsy with a histologic evaluation of fat deposition within hepatocytes [1,6,7].

Training as a radiologist includes understanding how liver size can change over time, and it is important to ensure liver spans are being measured correctly [13,14]. Recognizing the normal values that change with age and growth is crucial for an accurate evaluation. Organ size is crucial for the detection of abnormalities in disease [12]. The incorporation of imaging along with laboratory assessments is appropriate for the diagnosis and staging of MASLD in pediatrics. The use of an invasive liver biopsy will remain important in some cases; however, a combined initial approach of non-invasive testing and imaging will be the most beneficial [20].

Beginning with the initial description of steatohepatitis in obese children by Moran et al. in 1983, the prevalence of MASLD has continued to rise, necessitating the development of accessible and non-invasive options for early disease detection and treatment [4,23]. Rates will continue to rise with the obesity pandemic and early detection is vital for the prevention of severe disease and complications [2].

### Limitations

There are several limitations to the present study. This study was conducted at a single institution, including children seen at three separate pediatric clinics within the same metropolitan area. Therefore, the population of the study may not be generalizable to other geographic areas for children undergoing an ultrasound for hepatomegaly or increased liver enzymes. Further investigations, including multi-site studies, may strengthen these results with larger variability in the population. The use of ultrasound to determine hepatomegaly and increased echogenicity with a fellowship-trained pediatric radiologist review is a good test for MASLD and what we used as a gold standard for our study. However, follow up or confirmatory imaging with MRI, US, or CT scans with a focus on elastography during MRI to detect fibrosis would be helpful as well for confirmation. Splenomegaly has been seen in cases of MASLD, and complete abdominal ultrasound imaging may better determine any correlation of MASLD with splenomegaly.

## 5. Conclusions

Ultrasound remains a safe and readily available imaging modality that can help with the early detection of MASLD in pediatric patients. The use of laboratory data, including the hepatic function panel, is important but should be combined with imaging to increase detection rates for MASLD. Future research into other non-invasive imaging modalities, such as attenuation imaging and elastography, may reduce the need for invasive liver biopsy and can be used in conjunction with ultrasound and laboratory assessments as initial testing.

This study brings attention to the need for safer, more effective screening methods for the diagnosis of MASLD in the pediatric population. Although it is commonly used, there are still several children with a missed diagnosis. It is crucial to screen for MASLD in children, with the goal of identifying its presence before disease progression. These results will help to further emphasize the importance of observing proper age-based values for liver span with consideration of obtaining ultrasounds even with normal liver function tests as it is multifactorial for many young children.

## Figures and Tables

**Table 1 diagnostics-14-01652-t001:** Demographic characteristics of study participants.

	Overall (*n* = 134)
Age (months), mean (SD)	141.2 (53.45)
Age (months), *n* (%)	
>24–60	15 (11.2)
>60–144	49 (36.6)
>144–216	68 (50.7)
>216	2 (1.5)
Gender	
Male	77 (57.5)
Female	57 (42.5)
Race and Ethnicity	
White/Caucasian	46 (34.3)
Black/African American	43 (32.1)
Hispanic	28 (20.9)
Asian	6 (4.5)
Mixed Race	1 (0.8)
Other	10 (7.5)
BMI z-score, mean (SD) ^1^	1.27 (1.59)
BMI z score, *n* (%) ^1^	
<−1.2	10 (7.6)
−1.2–1.5	46 (35.1)
>1.5	75 (57.3)
ALT (U/L), mean (SD) ^1^	72.7 (140.00)
ALT (U/L), median (25, 75th percentiles) ^1^	28 (16, 61)
ALT (U/L) > 95th percentile, *n* (%) ^1^	74 (56.1)
ALT (U/L) > 95th percentile males, *n* (%)	54 (70.1)
ALT (U/L) > 95th percentile females, *n* (%) ^1^	20 (36.4)
ALT (U/L) above 2× ULN, *n* (%) ^1^	41 (31.1)
ALT (U/L) above 2× ULN males, *n* (%)	31 (40.3)
ALT (U/L) above 2× ULN females, *n* (%) ^1^	10 (18.2)
Diagnosis, *n* (%)	
Hepatomegaly	50 (37.3)
Abnormal Liver Enzymes	46 (34.3)
Acanthosis Nigricans	42 (31.3)
Abdominal Pain	62 (46.3)
Sleep Apnea	9 (6.7)
Other	36 (26.9)

^1^ Two missing observations.

**Table 2 diagnostics-14-01652-t002:** Hepatomegaly with ALT and race and ethnicity.

	Overall (*n* = 134)	Normal Liver Size (*n* = 60)	Hepatomegaly (*n* = 74)	*p* Value
**ALT > 95%**				<0.001
No	59 (43.9)	37 (62.1)	22 (29.7)	
Yes	75 (56.1)	23 (37.9)	52 (70.3)	
**Race and Ethnicity**				0.515
White/Caucasian	46 (34.3)	22 (36.7)	24 (32.4)	
Black/African American	43 (32.1)	21 (35.0)	22 (29.7)	
Hispanic	28 (20.9)	9 (15.0)	19 (25.7)	
Asian	6 (4.5)	4 (6.7)	2 (2.7)	
Mixed Race	1 (0.8)	0 (0.0)	1 (1.4)	
Other	10 (7.5)	4 (6.7)	6 (8.1)	

**Table 3 diagnostics-14-01652-t003:** Echogenicity with ALT and race and ethnicity.

	Overall (*n* = 134)	Normal Echogenicity (*n* = 72)	Increased Echogenicity (*n* = 62)	*p* Value
**ALT > 95%**				<0.001
No	59 (43.9)	45 (62.0)	14 (22.6)	
Yes	75 (56.1)	27 (38.0)	48 (77.4)	
**Race and Ethnicity**				0.018
White/Caucasian	46 (34.3)	30 (41.7)	16 (25.8)	
Black/African American	43 (32.1)	27 (37.5)	16 (25.8)	
Hispanic	28 (20.9)	9 (12.5)	19 (30.7)	
Asian	6 (4.5)	3 (4.2)	3 (4.8)	
Mixed Race	1 (0.8)	0 (0)	1 (1.6)	
Other	10 (7.5)	3 (4.2)	7 (11.3)	

**Table 4 diagnostics-14-01652-t004:** Agreement and performance measures of initial radiology read compared to read performed by the fellowship-trained pediatric radiologist (gold standard) for the diagnosis of hepatomegaly.

	Prevalence of Hepatomegaly	Correctly Classified *n* (%)	True Positive *n* (%)	False Positive *n* (%)	True Negative *n* (%)	False Negative *n* (%)	Sensitivity (95% CI)	Specificity (95% CI)	PPV (95% CI)	NPV (95% CI)
Overall (*n* = 134)	55.2%	102 (76.1%)	48 (35.8)	6 (4.5)	54 (40.3)	26 (19.4)	64.9% (52.9, 75.6%)	90.0% (79.5, 96.2%)	88.9% (77.4, 95.8%)	67.5% (56.1, 77.6%)
Provider specialty										
Adult specialist (*n* = 64)	54.7%	46 (71.9)	18 (28.1)	1 (1.6)	28 (43.8)	17 (26.6)	51.4% (34.0, 68.6%)	96.6% (82.2, 99.9%)	94.7% (74.0, 99.9%)	62.2% (46.5, 76.2%)
Pediatric specialist (*n* = 61)	52.5%	49 (80.3)	24 (39.3)	4 (6.6)	25 (41.0)	8 (13.1)	75.0% (56.6, 88.5%)	86.2% (68.3, 96.1%)	85.7% (67.3, 96.0%)	75.8% (57.7, 88.9%)
Unknown (*n* = 9)	77.8%	7 (77.8)	6 (66.7)	1 (11.1)	1 (11.1)	1 (11.1)	85.7% (42.1, 99.6%)	50.0% (1.3, 98.7%)	85.7% (42.1, 99.6%)	50.0% (1.3, 98.7%)
Males (*n* = 77)	63.6%	60 (77.9)	37 (48.1)	5 (6.5)	23 (29.9)	12 (15.6)	75.5% (61.1, 86.7%)	82.1% (63.1, 93.9%)	88.1% (74.4, 96.0%)	65.7% (47.8, 80.9%)
Females (*n* = 57)	43.9%	42 (73.7)	11 (19.3)	1 (1.8)	31 (54.4)	14 (24.6)	44.0% (24.4, 65.1%)	96.9% (83.8, 99.9%)	91.7 (61.5, 99.8%)	68.9% (53.4, 81.8%)
Race and ethnicity										
White (*n* = 46)	52.2%	32 (69.6)	13 (28.7)	3 (6.5)	19 (41.3)	11 (23.9)	54.2% (32.8, 74.4%)	86.4% (65.1, 97.1%)	81.2% (54.4, 96.0%)	63.3% (43.9, 80.1%)
Black (*n* = 43)	51.2%	33 (76.7)	15 (34.9)	3 (7.0)	18 (41.9)	7 (16.3)	68.2% (45.1, 86.1%)	85.7% (63.7, 97.0%)	83.3% (58.6, 96.4%)	72.0% (50.6, 87.9%)
Hispanic (*n* = 28)	67.9%	22 (76.6)	13 (46.4)	0 (0.0)	9 (32.1)	6 (21.4)	68.4% (43.4, 87.4%)	100.0% (66.4, 100.0%)	100.0% (75.3, 100.0%)	60.0% (32.3, 83.7%)
Other (*n* = 17)	52.9%	15 (88.2)	7 (41.2)	0 (0.0)	8 (47.1)	2 (11.8)	77.8% (40.0, 97.2%)	100.0% (63.1, 100.0%)	100.0% (59.0, 100.0%)	80.0% (44.4, 97.5%)
BMI z score										
<−1.2 (*n* = 10)	10.0%	10 (100.0)	1 (10.0)	0 (0.0)	9 (90.0)	0 (0.0)	100.0% (2.5, 100.0%)	100.0% (66.4, 100.0%)	100.0% (2.5, 100.0%)	100.0% (66.4, 100.0%)
−1.2–1.5 (*n* = 46)	43.5%	38 (82.6)	15 (32.6)	3 (6.5)	23 (50.0)	5 (10.9)	75.0% (50.9, 91.3%)	88.5% (69.8, 97.6%)	83.3% (58.6, 96.4%)	82.1% (63.1, 93.9%)
>1.5 (*n* = 75)	70.7%	52 (69.3)	32 (42.7)	2 (2.7)	20 (26.7)	21 (28.0)	60.4% (46.0, 73.5%)	90.9% (70.8, 98.9%)	94.1% (80.3, 99.3%)	48.8% (32.9, 64.9%)

**Table 5 diagnostics-14-01652-t005:** Agreement and performance measures of initial radiologist read compared to overread performed by fellowship-trained pediatric radiologist (gold standard) for the diagnosis of hepatic echogenicity.

	Prevalence of Echogenicity	Correctly Classified *n* (%)	True Positive *n* (%)	False Positive *n* (%)	True Negative *n* (%)	False Negative *n* (%)	Sensitivity (95% CI)	Specificity (95% CI)	PPV (95% CI)	NPV (95% CI)
Overall (*n* = 134)	46.3%	101 (75.4)	35 (26.1)	6 (4.5)	66 (49.3)	27 (20.2)	56.5% (43.3, 69.0%)	91.7% (82.7, 96.9%)	85.4% (70.8, 94.4%)	71.0% (60.6, 79.9%)
Provider specialty										
Adult specialist (*n* = 64)	56.2%	47 (73.4)	21 (32.8)	2 (3.1)	26 (40.6)	15 (23.4)	58.3% (40.8, 74.5%)	92.9% (76.5, 99.1%)	91.3% (72.0, 98.9%)	63.4% (46.9, 77.9%)
Pediatric specialist (*n* = 61)	31.1%	47 (77.1)	9 (14.8)	4 (6.6)	38 (62.3)	10 (16.4)	47.4% (24.4, 71.1%)	90.5% (77.4, 97.3%)	69.2% (38.6, 90.9%)	79.2% (65.0, 89.5%)
Unknown (*n* = 9)	77.8%	7 (77.8)	5 (55.6)	0 (0)	2 (22.2)	2 (22.2)	71.4% (29.0, 96.3%)	100.0% (15.8, 100.0%)	100.0% (47.8, 100.0%)	50.0% (6.8, 93.2%)
Males (*n* = 77)	53.2%	58 (75.3)	26 (33.8)	4 (5.2)	32 (41.6)	15 (19.5)	63.4% (46.9, 77.9%)	88.9% (73.9, 96.9%)	86.7% (69.3, 96.2%)	68.1% (52.9, 80.9%)
Females (*n* = 57)	36.8%	43 (75.4)	9 (15.8)	2 (3.5)	34 (59.7)	12 (21.1)	42.9% (21.8, 66.0%)	94.4% (81.3, 99.3%)	81.8% (48.2, 97.7%)	73.9% (58.9, 85.7%)
Race and ethnicity										
White (*n* = 46)	34.8%	39 (84.8)	11 (23.9)	2 (4.4)	28 (60.9)	5 (10.9)	68.8% (41.3, 89.0%)	93.3% (77.9, 99.2%)	84.6% (54.6, 98.1%)	84.8% (68.1, 94.9%)
Black (*n* = 43)	37.2%	30 (69.8)	5 (11.6)	2 (4.7)	25 (58.1)	11 (25.6)	31.2% (11.0, 58.7%)	92.6% (75.7, 99.1%)	71.4% (29.0, 96.3%)	69.4% (51.9, 83.7%)
Hispanic (*n* = 28)	67.9%	21 (75.0)	14 (50.0)	2 (7.1)	7 (25.0)	5 (17.9)	73.7% (48.8, 90.9%)	77.8% (40.0, 97.2%)	87.5% (61.7, 98.4%)	58.3% (27.7, 84.8%)
Other (*n* = 17)	64.7%	11 (64.7)	5 (29.4)	0 (0.0)	6 (35.3)	6 (35.3)	45.5% (16.7, 76.6%)	100.0% (54.1, 100.0%)	100.0% (47.8, 100.0%)	50.0% (21.1, 78.9%)
BMI z score										
<−1.2 (*n* = 10)	10.0%	10 (100.0)	1 (10.0)	0 (0.0)	9 (90.0)	0 (0.0)	100.0% (2.5, 100.0%)	100.0% (66.4, 100.0%)	100.0% (2.5, 100.0%)	100.0% (66.4, 100.0%)
−1.2–1.5 (*n* = 46)	26.1%	33 (71.7)	1 (2.2)	2 (4.4)	32 (69.6)	11 (23.9)	8.3% (0.2, 38.5%)	94.1% (80.3, 99.3%)	33.3% (0.8, 90.6%)	74.4% (58.8, 86.5%)
>1.5 (*n* = 75)	64.0%	56 (74.7)	33 (44.0)	4 (5.3)	23 (30.7)	15 (20.0)	68.8% (53.7, 81.3%)	85.2% (66.3, 95.8%)	89.2% (74.6, 97.0%)	60.5% (43.4, 76.0%)

## Data Availability

The original contributions presented in the study are included in the article/Appendix A. Further inquiries can be directed to the corresponding authors.

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
