# Peer review of "Efficacy of Ultrasound for the Detection of Possible Fatty Liver Disease in Children"

_diagnostics, 2024, doi:10.3390/diagnostics14151652_

Round 1

Reviewer 1 Report (Previous Reviewer 2)

Comments and Suggestions for Authors

"Efficacy of Ultrasound for Detection of Possible Fatty Liver 2

I have examined your study titled "Disease in Children" in detail. I have listed the points I found missing in the study in bullet points. The first is the emphasis on the method used in the abstract section. A paragraph regarding the article's organization should be added at the end of the Introduction section. I think that Table 4 and Table 5 are not interpreted sufficiently. It is possible to write the Conclusion section in more detail. I also believe the critical changes you made in the article have improved its quality.

Comments on the Quality of English Language

Spelling and grammatical errors in the article should be reviewed.

Author Response

Highlighted changes in GREEN in document

Comment 1: The first is the emphasis on the method used in the abstract section. 

Response 1: added an ending to say what we were actually studying 

Comment 2: A paragraph regarding the article's organization should be added at the end of the Introduction section.

Response 2: added a short paragraph to give an overview of paper

Comment 3: I think that Table 4 and Table 5 are not interpreted sufficiently. 

Response 3: this was to put the details to give statistical analysis, may be able to be better formatted to read easier in print but I was not able to adjust much. wanted to provide details on analysis though. 

Comment 4: It is possible to write the Conclusion section in more detail.

Response 4: Added another paragraph to highlight the importance of the study and our overall conclusions and observations.

Reviewer 2 Report (New Reviewer)

Comments and Suggestions for Authors

The paper by Lowry et al is well written, but the aim is not clear. The aim is  not even mentioned in the abstract. In the Introduction the  authors  mentioned two aims - prevalence of hepatomegaly in the pediatric population and to compare the diagnostic accuracy of these findings  compared to an overread performed by a fellowship trained pediatric radiologist. It is confusing as to what exactly a fellowship trained pediatric radiologist is and to what extent he/she is a gold standard.

In the result section only ALT of the biochemical parameters is listed, why is AST missing?

And finally, my main concern is the novelty of this paper and its contribution to the filed. The ultrasound is a routine technique for MASLD diagnosis and the experience of a radiologist is crucial everywhere. Also, there are papers on the prevalence of MASLD and hepatomegaly in pediatric population, and this single center retrospective study only shows situation in a small part of the USA, as the authors already mentioned in the Limitations/

Author Response

Comment 1: The paper by Lowry et al is well written, but the aim is not clear. The aim is  not even mentioned in the abstract.

Response 1: Aim added, highlighted in green (with response to reviewer 1)

Comment 2: In the Introduction the  authors  mentioned two aims - prevalence of hepatomegaly in the pediatric population and to compare the diagnostic accuracy of these findings  compared to an overread performed by a fellowship trained pediatric radiologist. It is confusing as to what exactly a fellowship trained pediatric radiologist is and to what extent he/she is a gold standard.

Response 2: community radiologists are reading several of these reports and are adult trained. There are specific age based criteria for liver span and therefore without pediatric training, the community radiologist may be less familiar with these values and may miss an enlarged liver. we highlight this to emphasize that pediatric training and reviewing more often pediatric imaging tests can lead to more accurate evaluation. This is why we wanted to highlight this as part of our study. 

Comment 3: In the result section only ALT of the biochemical parameters is listed, why is AST missing?

Response 3: We only report ALT as that is what is recommended for screening by NASPGHAN, not AST so ALT is what is reported upon as the standard for screening. 

Comment 4: And finally, my main concern is the novelty of this paper and its contribution to the filed. The ultrasound is a routine technique for MASLD diagnosis and the experience of a radiologist is crucial everywhere. Also, there are papers on the prevalence of MASLD and hepatomegaly in pediatric population, and this single center retrospective study only shows situation in a small part of the USA, as the authors already mentioned in the Limitations/

Response 4: Yes in North America, the recommendations still only include ALT so we want to highlight the importance of use of Ultrasound. it is not as routinely used by pediatricians so we are hoping to bring awareness. It was very well received when presented at Digestive Disease Week conference so we learned it was not as commonly understood in North America so we are hoping to help draw more attention to this matter. It is an important and increasing diagnosis as well in our patients. It could be more supported with other centers to do their own reviews yes of course. 

Round 2

Reviewer 1 Report (Previous Reviewer 2)

Comments and Suggestions for Authors

Congratulations on a successful revision. Words like we, our should not be used too much. Spelling and grammatical errors should be reviewed.

Comments on the Quality of English Language

.

Author Response

Comment 1:

Congratulations on a successful revision. Words like we, our should not be used too much. Spelling and grammatical errors should be reviewed.

Response 1:

Thank you! Changed out the verbage and reviewed grammar and spelling throughout. 

changes and new sentences/words highlighted in teal/light blue

Reviewer 2 Report (New Reviewer)

Comments and Suggestions for Authors

I suggest adding parts of Responses 2 and 4 into Discussion, to clarify some issues on novelty and significance of the manuscript.

Author Response

Comment 1: I suggest adding parts of Responses 2 and 4 into Discussion, to clarify some issues on novelty and significance of the manuscript

Response 1: I added a few sentences to highlight the importance/novelty with focus on ALT being the current recommended lab and only screening method, as well as emphasizing the importance of proper training and knowledge of appropriate liver sizes for accurate measurements and diagnoses. 

This manuscript is a resubmission of an earlier submission. The following is a list of the peer review reports and author responses from that submission.

Round 1

Reviewer 1 Report

Comments and Suggestions for Authors

title: ok

Abstract: improve the discussion section, including any limitations of the study

keywords: I suggest using the following NAFLD, Pediatric, Ultrasound, Fibrosis, hepatomegaly, MASLD.

Throughout the text: bibliographical references should normally be inserted before the "." symbol,  at the end of the sentence

Introduction: I suggest combining the first two sentences to be more concise and readable.

methods: the last paragraph of sector 2.2 is not well readable; please clarify what you mean by "the fellowship trained pediatric radiologist" (how many years of experience? how many tests performed each year?)

results and conclusions: nothing to change

Author Response

title: ok

Abstract: improve the discussion section, including any limitations of the study

  • Had to edit it down more due to character count in template

keywords: I suggest using the following NAFLD, Pediatric, Ultrasound, Fibrosis, hepatomegaly, MASLD.

  • Edited keywords but we don’t do elastography so didn’t add fibrosis and kept echogenicity as it is measured in our study

Throughout the text: bibliographical references should normally be inserted before the "." symbol,  at the end of the sentence

  • Edited this throughout the paper (not going to highlight each reference)

Introduction: I suggest combining the first two sentences to be more concise and readable.

  • Edited to combine

methods: the last paragraph of sector 2.2 is not well readable; please clarify what you mean by "the fellowship trained pediatric radiologist" (how many years of experience? how many tests performed each year?)

  • I don’t have this data for tests but just needed to clarify there is additional training for pediatric fellowship

results and conclusions: nothing to change

Reviewer 2 Report

Comments and Suggestions for Authors

I examined your study titled "Efficacy of Ultrasound for Detection of Possible Fatty Liver Disease in Children" in detail. To improve the quality of your article, I have listed the missing points in bullet points. In the abstract section, where and how the data was collected and the results of the study are discussed. However, the method used in the study was not mentioned. The method has been added to the abstract section. At the end of the Introduction section, a section about the article's innovations and contributions to the literature should be added, and then a paragraph about the organization of the article should be added. In this section, a literature review on the subject should also be made. The difference between this study from similar ones should be explained. Words like "we" were used a lot in the study. Title 2.4 The statistical analysis section is one of the most important steps of the study. The statistical methods used here should be explained in more detail; the statistical methods used in this section are not explained in detail. The relevant part should be expanded, including the equations used here. I barely understood the values in the tables presented in the application section, but it is difficult for someone who is not an expert in this field to understand the values in these tables. Tables are always expressed by writing numerical values. These values are already available in the tables, rather it is important to explain these values. The conclusion section should be expanded. I would like to point out that the rest of the work is well organized.

Comments on the Quality of English Language

Spelling and grammatical errors in the article should be reviewed.

Author Response

I examined your study titled "Efficacy of Ultrasound for Detection of Possible Fatty Liver Disease in Children" in detail. To improve the quality of your article, I have listed the missing points in bullet points.

 In the abstract section, where and how the data was collected and the results of the study are discussed. However, the method used in the study was not mentioned. The method has been added to the abstract section.

  • We talked briefly about the methods of reviwing the ultrasound which was the primary thing

At the end of the Introduction section, a section about the article's innovations and contributions to the literature should be added, and then a paragraph about the organization of the article should be added. In this section, a literature review on the subject should also be made. The difference between this study from similar ones should be explained. Words like "we" were used a lot in the study.

  • In the highlighted last paragraph it explains what has previously been looked at and what our study adds. The introducion has the literature review throughout as we discuss the importance of this topic and what we are trying to add to create more awareness.

 Title 2.4 The statistical analysis section is one of the most important steps of the study. The statistical methods used here should be explained in more detail; the statistical methods used in this section are not explained in detail. The relevant part should be expanded, including the equations used here. I barely understood the values in the tables presented in the application section, but it is difficult for someone who is not an expert in this field to understand the values in these tables. Tables are always expressed by writing numerical values. These values are already available in the tables, rather it is important to explain these values. The conclusion section should be expanded. I would like to point out that the rest of the work is well organized.

  • The tables have total number (n) and Confidence interval and %. I didn’t expand to add those words since it seems standard but I can always add those details in. it was written what we did do
  • I had put more details into the discussion to make the conclusion short and concise